# Investigation of the Maturity Evaluation Indicator of Honey in Natural Ripening Process: The Case of Rape Honey

**DOI:** 10.3390/foods10112882

**Published:** 2021-11-22

**Authors:** Guo-Zhi Zhang, Jing Tian, Yan-Zheng Zhang, Shan-Shan Li, Huo-Qing Zheng, Fu-Liang Hu

**Affiliations:** College of Animal Sciences, Zhejiang University, No. 866, Yuhangtang Road, Xihu District, Hangzhou 310058, China; zhangguozhi@zju.edu.cn (G.-Z.Z.); tj_1224@126.com (J.T.); 21417023@zju.edu.cn (Y.-Z.Z.); lishanshan@zju.edu.cn (S.-S.L.); hqzheng@zju.edu.cn (H.-Q.Z.)

**Keywords:** rape honey, maturity, antioxidant, volatile profile

## Abstract

Honey maturity, a critical factor for quality evaluation, is difficult to detect in the current industry research. The objective of this study was to explore the changes in the composition and find potential maturity indicators of rape honey at different maturity stages through evaluating physicochemical parameters (moisture, sugars, pH, electrical conductivity, total protein, total phenols, total flavonoids, proline, and enzyme activity), the antioxidant capacity, and volatile components. The relevant results are as follows: 1. As the maturity increased, the moisture, sucrose, and maltose content of rape honey gradually decreased, while the glucose, fructose, and total protein content gradually increased. The activities of diastase, invertase, and β-glucosidase showed a significant increase with the elevation of ripening days, and the activity of glucose oxidase reached the highest before completely capping. 2. The antioxidant capacity of honey increased with the increase in honey maturity. There is a significant and strong correlation between the bioactive components of rape honey and antioxidant capacity (*p* < 0.01, |r| > 0.857). 3. Thirty-five volatile components have been identified. Nonanal, benzaldehyde monomer, and benzaldehyde dimer can be used as potential indicators for the identification of honey maturity stages. Principal component analysis (PCA) based on antioxidant parameters and volatile components can identify the maturity of honey.

## 1. Introduction

Honey, the natural sweet substance, is produced by honey bees from plant nectar or from plant secretions or honeydew of plant-sucking insects (*Hemiptera*), which the bees collect, transform by mixing with their secretions, and place them into the cells of honey comb to mature [1,2]. Driven by profit, some beekeepers in China harvest raw honey every 2 or 3 days in order to increase honey production. Then the dilute honey is artificially dehydrated to meet relevant honey standards, which is defined as honey fraud by Apimondia [3]. The maturity of honey has a great impact on its quality. Immature honey (uncapped honey) restricts the healthy development of the honey market. Due to the lack of methods to distinguish between mature honey and immature honey, mature honey cannot be of high quality and price, and the proportion of mature honey in China in total honey production is still low.

In recent years, there has been a rising trend in the research on mature honey. Proline content was once considered as an indicator of mature honey and honey with a proline content of less than 180 mg/kg was considered as immature honey [4], but Zhang et al. [5] found there is no correlation between immature honey and its proline content. A series of physical and chemical reactions in honey occur for the ripening process. The composition of immature honey is more similar to nectar due to the short processing time by bees. Gil et al. [6] found that rosemary nectar contains kaempferol-3-sophoroside and quercetin-3-sophoroside, but these two glycosides could not be detected in rosemary honey. As a result, the author speculated that the bee enzymes hydrolyzed glycosides in nectar. Truchado et al. [7] revealed that immature honey and mature honey have significant differences in the composition of glycosides and phenols, and immature honey has more glycosides while mature honey contains more phenols. The protein content can well represent the natural maturity of honey [8]. Ma et al. [9] successfully identified mature acacia honey and immature acacia honey by determining 18 physicochemical parameters including total phenols, total protein, and enzymes activity, combined with chemometric profiling. Gismondi et al. [10] demonstrated that acacia honey has more secondary metabolites and distinct SDS-PAGE (sodium dodecyl sulfate-polyacrylamide gel electrophoresis) protein profiles than acacia nectar. The antibacterial activity and antioxidant capacity of acacia honey were significantly better than acacia nectar. Compared with immature honey, mature honey contains different phenolic components and has stronger antibacterial and antioxidant capacity [11]. The content of 9 phenolic compounds in mature honey and immature honey, such as gallic acid and caffeic acid, showed significant differences, which can be regarded as an important reference marker to discriminate mature honey from immature honey [12]. Sun et al. [13] researched the changes in fatty acids during the transition from immature honey to mature honey by ultrahigh-performance liquid chromatography-mass spectrometry (UPLC-MS/MS) and gas chromatography-mass spectrometry (GC-MS). Finally, decenedioic acid, the bee-derived component, was confirmed as a potential indicator for distinguishing mature honey from immature honey.

Generally, volatile compounds in honey could be originated from the plants, from honey bees, from honeydew of plant-sucking insects, from the transformation of plant compounds by honey bees, from heating or handling during honey processing and storage, or from microbial or environmental contamination [14]. Furthermore, some authors associate the volatile components with honey maturity. Naef et al. [15] determined the volatile components in linden nectar, stomach, and mature honey, and exhibited that nectar and stomach honey contain a variety of aldehydes, which are probably converted into corresponding acids in the process of ripening. In addition, the author also detected two queen pheromones derived from bees in mature honey. Wang et al. [16] explored the correlation between volatile components in buckwheat honey and maturity, and the result presented that below 40 Baume degree (°Bé) samples contained more esters and alcohols, and above 40 °Bé samples had the higher content of aldehydes and acids.

Notwithstanding that some progress has been accomplished in the research of mature honey, the distinction between mature honey and immature honey remains to be well understood. To establish a mature honey identification system, the relevant indicators of mature honey must primarily be clarified, and the research on the natural ripening process of honey is helpful to the formulation of relevant standards for mature honey. Considering all these aspects, we determined the physicochemical parameters, antioxidant capacity, and volatile components of rape honey at different ripening stages, hoping to explore the changes in honey’s composition during the honey ripening process and screen out potential honey maturity indicators.

## 2. Materials and Methods

### 2.1. Honey Samples

Raw honey samples at different maturity stages were collected by 500 g rectangular grid comb honey boxes from five strong *Apis mellifera* L. colonies in Taizhou city, Zhejiang province, during the rape flower season from 29 March to 13 April 2019. Before the formal honey sampling, the comb foundation of the comb honey box must be raised by bees, and the honey stored in it must be removed. The comb honey box was put into the colony in the morning. Then, we took a comb honey box on the 1, 2, 3, 4, 5, 6, 7, 8, 9, 10, 11, 12, 14, and 16 days, and calculated the capping ratio of comb honey. We mixed the honey samples of the same ripening day in 5 bee colonies into one sample, and the botanical origin of honey was assessed by the beekeepers and then confirmed by melissopalynological analysis [17]. 14 different ripening days honey samples were collected, and were stored at −20 °C in the dark immediately for further analysis. 

### 2.2. Main Reagents and Equipments

#### 2.2.1. Main Reagents

All chemicals and reagents used were of analytical grade. Methanol and acetonitrile were purchased from Merck Co., Inc. (Darmstadt, Germany). Fructose, glucose, sucrose, and maltose standards were obtained from Shanghai Yuanye Biotechnology Co., Ltd. (Shanghai, China). 3,5-Dinitrosalicylic acid and indigo carmine were purchased from Shanghai Dibo Biotechnology Co., Ltd. (Shanghai, China). 4-nitrophenol and gallic acid were from Shanghai Macklin Biochemical Co., Ltd. (Shanghai, China). DPPH and TPTZ were purchased from Sigma-Aldrich (St. Louis, MI, USA). Proline and rutin were obtained from Beijing Bailingwei Technology Co., Ltd. (Beijing, China).

#### 2.2.2. Main Equipments

Conductometer DDS-11A: Shanghai INESA Scientific Instrument Co., Ltd. (Shanghai, China). pH meter EL20: Mettler-Toledo Instruments (Shanghai) Co., Ltd. (Shanghai, China). Shimadzu spectrophotometer UV-2550: Shimadzu Co., Ltd. (Kyoto, Japan). Agilent 1260 liquid chromatograph: Agilent Technologies, Palo Alto, CA, USA. Multiskan Sky: Thermo Scientific (Waltham, MA, USA). Agilent 490 gas chromatograph: Agilent Technologies, Palo Alto, CA, USA. IMS instrument: FlavourSpec^®^, Gesellschaft für Analytische Sensorsysteme mbH (Dortmund, Germany).

### 2.3. Physicochemical Parameters of Honey Samples

The moisture, electrical conductivity, and pH were measured according to the method reported by de Almeida-Muradian et al. [18]. At 20 °C, the refractometer was used to determine the moisture content of the honey samples. The electrical conductivity of a honey solution 20 g/100 g (dry matter basis) in pure water was measured at 20 °C in a conductometer. 10 g honey sample was dissolved in 75 mL distilled water, and the pH value was measured with a pH meter. The diastase activity was analysed by the spectrophotometric method [19]. The starch solution and honey buffer solution were incubated in the water bath pot at 40 °C. Diastase number (DN: 1% (*w*/*v*) starch solution hydrolyzed per g honey in 1 h) can be used to denote the amount of activity.

The measurement of sugars in the honey samples was determined by a high-performance liquid chromatograph (Agilent 1260) equipped with a refractive index detector, using the procedure described by Bentabol-Manzanares et al. [20]. For the total protein content, the extraction of protein in honey referred to the method of Zhang et al. [21]; the protein content was determined by the Bradford method [22], and bovine serum albumin was used as the standard.

The β-glucosidase activity was examined as described in Low et al. [23]. 1.0 g of honey was dissolved in 10 mL of disodium hydrogen phosphate-citrate buffer (pH 5.0). Then, 0.5 mL of honey solution and 0.5 mL of 30 mM 4-nitrophenyl-β-*D*-glucopyranoside solution were mixed and incubated in a water bath pot at 37 °C for 1.5 h. After incubation, 2.5 mL Na_2_CO_3_ solution (1 M) was added to stop the reaction. The absorbance was measured at 400 nm. The β-glucosidase activity in honey is expressed by the international units (U/g). The determination of invertase activity was performed as described in Yuan et al. [24] with minor modifications. The mixed solution of honey and sucrose was incubated in a water bath pot at 45 °C for 1 h. 1 mL of the solution before and after conversion were mixed with 2 mL 3,5-dinitrosalicylic acid solution. After heating 3 min in boiling water and cooling in running water for 5 min, the mixed solution was adjusted with distilled water to 25 mL and shook evenly. The absorbance was measured at a wavelength of 540 nm. The invertase activity (mg/g·h) was represented as the number of milligrams of sucrose converted into monosaccharides within 1 g of honey in 1 h. The detection of glucose oxidase in honey was carried out using the protocol of Ma et al. [9]. The glucose oxidase activity in honey is defined as the μg of hydrogen peroxide produced by the glucose oxidase in honey oxidizing glucose at 37 °C for 0.5 h.

### 2.4. Determination of Total Phenolic Content (TPC), Total Flavonoids Content (TFC) and Proline Content

TPC analysis was performed using the Folin-Ciocalteu method [25] with some modifications. In brief, 0.5 mL of honey solution (0.2 g/mL) and 2.5 mL of Folin-Ciocalteu reagent (0.2 M) were mixed in a centrifuge tube. The mixture was left to stand for 5 min in the dark, followed by the addition of 2 mL of Na_2_CO_3_ solution (0.1 g/mL) and stirred. After 2 h of reaction at room temperature without light, the absorbance was detected at 725 nm. The standard curve was produced using gallic acid (0.04–0.28 mg/mL).

The determination of TFC was achieved as described in Zhang et al. [17] and Tang et al. [26] with the following modifications: 1 mL honey solution (0.2 g/mL) was mixed with 0.3 mL NaNO_2_ (15 g/100 mL). At 6 min and 12 min, 0.3 mL Al(NO_3_)_3_ (10%) and 4 mL NaOH (1 M) were added, respectively. Finally, the above solution was diluted with 50% ethanol to 10 mL, homogenized and allowed to stand for 15 min. The absorbance was determinated at 510 nm in a spectrophotometer. The TFC was deduced from a standard curve and calculated in mg quercetin equivalent (mg QE/100 g).

To estimate the proline content of honey samples, the method of Meda et al. [25] was employed with modifications. 1 mL aqueous solution of honey (0.05 g/mL) was mixed with 250 μL formic acid (80%), 1 mL 3% ninhydrin solution (3 g ninhydrin in 100 mL ethylene glycol monomethylether) and oscillated for 15 min. The mixture was put in a boiling water bath for 15 min and then incubated in a 70 °C bath for 10 min. A 5 mL solution of 50% 2-propanol in distilled water was added and then the final solution was cooled and stood for 45 min. Water was regarded as the blank, and the absorbance of samples was performed at 510 nm. The standard curve was defined by known concentrations of proline standard, ranging between 5 and 25 μg/mL.

### 2.5. Determination of Antioxidant Capacity

#### 2.5.1. DPPH (2,2-Diphenyl-1-picrylhydrazyl) Radical Scavenging Ability

The DPPH assay was monitored as described in Turkmen et al. [27] with some modifications. Honey solution (0.5 g/mL) was prepared with distilled water, and then it was diluted to various concentrations. 100 μL samples were mixed with 100 μL of DPPH in methanol (50 μg/mL) in wells of a 96-well plate. Appropriate blanks of honey solution and of DPPH reagent (to correct for color of the honey solution) were run. The plate was left at room temperature in the dark for 30 min. Absorbance was measured at 517 nm and results were presented as a percentage of EC50 (the concentration reducing the original absorbance of DPPH by 50%).

#### 2.5.2. Ferric Reducing/Antioxidant Power (FRAP) Assay

FRAP assay was assessed according to Biluca et al. [28] with slight modifications, using an aqueous solution of honey (0.1 g/mL). FRAP reagent was produced by mixing 300 mM sodium acetate buffer (pH 3.6), 10 mM tripyridyltriazine (TPTZ) in 40 mM HCl, and 20 mM FeCl_3_ in the volume ratio 10:1:1. 0.5 mL honey solution and 5 mL FRAP reagent were vortex-mixed and placed in a 37 °C water bath for 10 min. The absorbance was measured at 593 nm, and results were expressed in Fe^2+^ equivalents (mmol Fe^2+^/100 g) with a standard curve between 0.2 and 1.0 mM.

#### 2.5.3. Reducing Power

The reducing power of honey was evaluated using the protocol reported before [29]. 1 mL honey solution (0.2 g/mL) was homogeneously mixed with 2.5 mL phosphate buffer (0.2 M, pH 6.6) and 2.5 mL 1% K_3_Fe(CN)_6_, and incubated at 50 °C for 20 min. After incubation, 2.5 mL of 10% trichloroacetic acid was added, and the mixture was centrifuged at 10,000 rpm for 10 min. 2.5 mL of the clear upper layer was mixed with 2.5 mL of deionized water and 0.5 mL of 0.1% FeCl_3_. Then, the reaction mixture was vortexed and stood for 10 min. The absorbance was read at 700 nm and the standard curve was plotted for rutin within the concentration range from 0.04 to 0.4 mg/mL.

### 2.6. Analysis of Volatile Constituents by Headspace Gas Chromatography-Ion Mobility Spectrometry (HS-GC-IMS)

We selected samples with more significant physicochemical properties and antioxidant capacity, that is, samples on days 1, 2, 3, 5, 7, 9, 12, 14, and 16 for the determination of volatile constituents. The analyses were monitored on a GC-IMS, based on an Agilent 490 gas chromatograph, equipped with an automatic headspace sample injector, using the method of Wang et al. [30] with minor modifications. 1.0 g honey sample was placed in a 20 mL headspace vial, then the sample was incubated at 500 rpm and at 40 °C for 15 min. Subsequently, 500 μL of headspace was automatically injected in splitless mode by means of a heated syringe at 85 °C. N_2_ (purity ≥ 99.99%) was utilized as the carrier gas, and the sample was separated at 60 °C in the column. The carrier gas followed a set flow: 2 mL/min for 2 min, flow increased to 100 mL/min at 20 min, and held for 10 min. The drift tube was used N_2_ as the drift gas (the flow rate of 150 mL/min) and maintained at a constant voltage of 500 V/cm and a temperature of 45 °C. Each spectrum was scanned 16 times on average with a repetition frequency of 30 ms. The analysis was carried out in duplicate and expressed as means.

### 2.7. Statistical Data Analysis

Except for the analysis of volatile constituents, other results were conducted in triplicate and expressed as means ± standard deviation. Significant differences were determined using Tukey’s honestly significant difference test, and *p* < 0.05 was considered to be statistically significant. Correlation analysis was achieved using SPSS 23.0 software (IBM SPSS Statistics; Chicago, IL, USA). Correlation coefficients (r) were calculated by Pearson’s correlations between antioxidant components and antioxidant activities. Laboratory Analytical Viewer (LAV), 3 plug-ins (reporter plugin, gallery plot plug-in, and dynamic principal component analysis (PCA) plug-in), and GC-IMS Library Search software were used for analyzing the data of volatile constituents [31].

## 3. Results

### 3.1. Analysis of Physicochemical Properties

The results of the melissopalynological analysis revealed that rape honey samples had a percentage of pollen grains above the minimum of 45%. It showed that honey samples in our research were monofloral honeys. The physicochemical and chemical properties of honey samples with different maturity stages were shown in Table 1. With the increase of ripening days, the water content of honey showed a significant downward trend, from 26.9% to 18.5% (*p* < 0.05). The capping of honey started on the 5th day (2.3 ± 0.2%), the capping ratio reached 69 ± 6% on the 10th day, and 100% on the 14th day.

We found that with the increase in honey maturity, the glucose and fructose content showed a generally rising trend, while the sucrose and maltose content showed an overall decline trend. The glucose content of honey samples fluctuated between 34 ± 5% to 36 ± 3% from the 4th day to the 12th day, and the fructose content fluctuated between 29 ± 2% to 30 ± 2% from the 4th day to the 7th day. The content of sucrose and maltose was significantly reduced during the honey ripening process (*p* < 0.05).

The pH value of honey we tested varied from 3.54 to 3.72 and the electrical conductivity in honey was in the range 152–179 μS/cm. During the ripening process, the protein content of samples grew from 6.7 mg/g to 8.8 mg/g. 

As the maturity increased, the DN, invertase activity, and β-glucosidase activity exhibited an escalating trend, growing from 16 to 29, from 44 mg/g·h to 189 mg/g·h, and from 0.85 U/g to 1.41 U/g, respectively. The glucose oxidase activity of rape honey samples ranged from 133 μg/g·0.5 h to 353 μg/g·0.5 h. During the first 12 days (except for fluctuations on the 8th day), the glucose oxidase activity increased steadily. The highest activity was attained before absolutely capping and decreased on the 14th and 16th days.

### 3.2. Antioxidant Analysis and Proline Content

We detected the TPC, TFC, proline content, DPPH radical scavenging capacity, FRAP and the reducing capacity of rape honey samples at different maturity stages. The relevant results were shown in Table 2. The TPC of rape honey displayed an overall uptrend with the increase of ripening days, from 23.8 mg GAE/100 g to 32 mg GAE/100 g, and the TPC of capped honey samples (14-day and 16-day) were significantly higher than uncapped (1–4 days). The variation trend of TFC was similar to that of TPC, with a slight fluctuation in the overall increase, from 24 mg QE/100 g to 35 mg QE/100 g. The content of proline increased significantly from 212 mg/kg to 318 mg/kg.

With the increase in honey maturity, the EC50 decreased from 0.63 g/mL to 0.31 g/mL, the FRAP increased from 0.74 mmol Fe^2+^/100 g to 2.3 mmol Fe^2+^/100 g, and the reducing power augmented from 83 mg/100 g to 118 mg/100 g, which suggesting the antioxidant capacity of rape honey continues to increase.

Table 3 presented the correlations among the analysis of bioactive components and antioxidant capacity. There was a significant strong correlation between all parameters (*p* < 0.01, |r| > 0.857). The TPC had strong positive correlation with TFC, proline content, FRAP, and reducing power (r = 0.866, r = 0.882, r = 0.962, r = 0.954, respectively), but negative correlation with DPPH radical scavenging capacity (r = −0.950). There was a strong correlation between TFC and proline content (r = 0.971), FRAP (r = 0.860), reducing power (r = 0.915), and DPPH radical scavenging capacity (r = −0.909). Proline content had a strong positive correlation with FRAP (r = 0.857) and reducing power (r = 0.916), but it was negatively correlated with DPPH radical scavenging capacity (r = −0.919). DPPH radical scavenging capacity had a strong correlation with FRAP (r = −0.971) and reducing power (r = 0.963). The correlation between FRAP and reducing power was 0.969. These three methods for evaluating the antioxidant capacity of honey have similar results.

### 3.3. Analysis of Volatile Profiles

We normalized the ion migration time and the position of the reactive ion peak (RIP) to obtain a two-dimensional top view of the HS-GC-IMS (Figure 1). Each point on either side of the RIP represented a volatile organic compound and the color of the point expressed material concentration, where white indicated low concentration, red indicated high concentration, and a darker color indicated a greater concentration. A total of 53 volatile substances were detected, 35 of which were identified by the National Institute of Standards and Technology (NIST) and IMS databases (Table 4), including 8 aldehydes, 8 ketones, 5 alcohols, 8 esters, 3 acids, a furan, a disulfide, and an amine compound.

We used the Gallery Plot plug-in to draw fingerprints of volatile substances (Figure 2), and compared the differences in volatile organic compounds between different samples intuitively. Combining Figure 2 and Table 4, with the prolonging of ripening time, the content of 3-octanol, propyl butanoate, hexanal dimer, n-propyl acetate, 1-butanol monomer, pentanal, butyl acetate and hexanal monomer decreased initially but stabilized afterwards, while the content of 1-butanol dimer, 2-butanone dimer, acetone, and methyl isobutyl ketone monomer sustainably accumulated in the samples of the first 3 days or the first 5 days, and then decreased. The content of nonanal, 2-methylbutanoic acid dimer, isovaleric acid, (Z)-3-hexen-1-ol dimer, acetic acid, furfural, ethyl 2-methylbutyrate, and 2-hexanone generally increased during the ripening process.

The PCA results of 41 indicators (antioxidant and 35 of identified volatile components) for 9 honey samples at different maturity stages are shown in Figure 3. The two principal components explained 83.66% of the total variance, the first principal component (PC1) explained 61.18%, and the second principal component (PC2) explained 22.48%. As shown in the PCA score plot (Figure 3A), the samples studied were discriminated into three different groups which corresponded with their ripening stages. 1-day and 2-day honey samples were scattered in the II quadrant (early maturity stage). 3-day and 5-day honey samples were dispersed in the circle of medium maturity stage (in Ш quadrant). The honey samples over 7 days were classified as late maturity stage. The score plot (Figure 3B) indicates the position of the 41 indicators. The samples in the early maturity stage seemed characterized by most alcohols (3-octanol, pentan-1-ol, 1-butanol-monomer), aldehydes (hexanal dimer, hexanal monomer, pentanal), and butyl acetate (Figure 3B). Methyl isobutyl ketone (dimer and monomer) dominated in the samples of medium maturity stage. The location of honey samples at the late maturity stage was positively correlated with antioxidant parameters (TPC, TFC, FRAP, reducing power), benzaldehyde (dimer and monomer), and isopropyl acetate, which indicated these samples had a relatively high content of antioxidant components and better antioxidant capacity.

## 4. Discussion

There are currently more than 180 ingredients in honey, some of which are produced during the ripening process [32,33]. In order to find out the indicators of honey maturity, we collected rape honey of different maturity stages, and conducted experiments from three aspects: physicochemical parameters, antioxidant activity and volatile components.

### 4.1. Effects of Physicochemical Indexes on Evaluation of Honey Maturity

The maturity of honey intensely affects its physicochemical properties. In the colony, worker bees repeatedly brew nectar, and finally cap the mature honey with beeswax. During the ripening process, a series of physical and chemical changes have occurred [34]. The sealing time of honey is mainly affected by the vigor of the bee colony and climatic conditions. In our research, it took 14 days for the honey to reach full capping. Water content is an important parameter in honey quality and is related to honey fermentation, maturity and crystallization [35]. During the ripening process of honey, the moisture keeps decreasing [36]. The Codex Alimentarius [1] and European standards [2] stipulated that the water content of honey should be less than 20%. In our study, the water content of the samples became 20% by day 7 and only decreased further by day 12. The moisture of capped honey samples (14-day and 16-day) was significantly lower than uncapped (1 to 4-day).

Sugars in honey account for 95% to 99% of its dry weight, mainly fructose and glucose [37,38]. The fluctuation between the fructose and glucose content in 4–7 days honey was likely related to various enzyme activities and the addition of new honey. The content of monosaccharides (fructose and glucose) in all honey samples was not less than 60%. There were no obvious differences (*p* > 0.05) in the monosaccharide during the honey ripening process. The sucrose content of all honey samples was in good consistency with the specification of less than 5% [1,2].

A previous study demonstrated that the crystallization rate of honey depended on the ratio of fructose and glucose (F/G). When F/G ≤ 1.14, it means that honey is prone to crystallize quickly; when 1.3 < F/G < 1.58, it means that the crystallization rate is slow, but F/G > 1.58, honey will not crystallize [39]. The F/G in our samples ranged from 0.83 to 0.93 (data not shown), which is consistent with the fact that rape honey is easy to crystallize.

The pH value influences the taste, stability and shelf life of honey [40]. The pH in our research was similar to those found by Bogdanov et al. [4]. Electrical conductivity is relevant to ash and acidity [39]. According to the Codex Alimentarius [1] and European standards [2], the allowable upper limit for electrical conductivity is 800 μS/cm. None of the tested honey was exceeded the upper limit value. The conductivity and pH of honey with different ripening days had no obvious change tendency, as described in the literature [9].

The protein content in honey is chiefly concerned with the ripening degree [41]. Bees constantly add their secretions into honey, including royal jelly protein and some enzymes. The enzyme activity in honey is an important parameter for honey quality control. We measured the activities of diastase, invertase, glucose oxidase and β-glucosidase in rape honey with different maturity stages. The DN in all honeys was accorded with the standards (not less than 8) [1,2]. Another study also proposed that the honey DN of ripening for one day was still greater than 8 [8]. Sucrose is decomposed into fructose and glucose by invertase, which is the key enzyme for the conversion of nectar to honey [42]. The invertase activity ranged from 44 ± 6 mg/g·h to 189 ± 19 mg/g·h, being within the norms of Bogdanov et al. [4], which was not less than 40 mg/g·h. Similar to the current assay, Yuan et al. [24] proposed that the invertase activity of immature rape honey (3–4 days) was 19.32–60.19 mg/g·h, which was lower than our samples (85–112 mg/g·h). Under aerobic conditions, glucose oxidase can specifically catalyze β-*D*-glucose to produce gluconic acid and hydrogen peroxide. Gluconic acid is the main acidic substance in honey, increasing the acidity of honey, while hydrogen peroxide is one of the main materials for kinds of honey to exert antibacterial activity [43]. Ye et al. [8] measured the glucose oxidase activity of rape honey from 1 to 30 days. Except for the fluctuation in the first three days, it showed an upward tendency in 4 to 30 days, from 60.30 μg/g·0.5 h to 129.50 μg/g·0.5 h. However, our study found that glucose oxidase activity decreased after complete capping. β-glucosidase can hydrolyze the non-reducing β-*D*-glucoside, and release β-*D*-glucose and the corresponding ligand. In the experience, β-glucosidase activity agreed with the results of Yi et al. [44], the β-glucosidase activity of 41 °Bé honey was higher than 37 °Bé and 39 °Bé.

With the exception of moisture, the physicochemical properties of all honey samples were in compliance with the relevant standards. In moisture, sucrose, maltose, diastase, invertase, glucose oxidase and β-glucosidase, there was a significant difference between capped honey samples (14-day and 16-day) and uncapped honey samples (1-day and 2-day).

### 4.2. Effects of Bioactive Constituents and Antioxidant Capacity on Evaluation of Honey Maturity

A previous study has indicated that the TPC and antioxidant capacity of mature honey was significantly higher than that of immature honey [11]. The TPC of honey samples was in line with the reported in the literature at 5.6–50 mg GAE/100 g [45,46]. Our data did not support the claim that honey with a proline content greater than 180 mg/kg was mature honey, because the proline content of our honey reached 212 ± 13 mg/kg on the first day. The proline content in rape honey was consistent with the previous study (142–466 mg/kg) [47].

Saxena et al. [48] reported the correlations between proline content, DPPH radical scavenging capacity and FRAP. The study of Costa et al. [49] indicated proline had a strong correlation with phenolic compounds and FRAP. The above was basically consistent with our research, indicating that the total phenols, total flavonoids and proline of rape honey with different ripening days had a strong correlation with its antioxidant capacity, which were important components of antioxidant capacity. 

### 4.3. Effects of Volatile Compounds on Evaluation of Honey Maturity

Volatile compounds in natural raw honey mainly originated from the nectar plant, and from the transformation of plant-derived compounds by bees [14]. HS-GC-IMS has high sensitivity, high resolution, and the analyzed samples require no pre-processing. It has been widely used in the field of food flavor analysis [50,51]. PCA is a common multivariate method of mathematical statistics, which identifies differences and associations between variables and samples by reducing the dimensionality of the data set [52]. It is generally believed that when the cumulative variance contribution rate reaches 60%, PCA is a more appropriate separation model [53]. A previous study has demonstrated that the number and concentration of alcohols, esters and terpenes in honey samples decreased through ripening processes [33]. But in our results, the concentration of 3 alcohols and 3 esters increased during ripening processes. Acetone is the highest content of volatile components in rape honey [54], but it was undulated and had no obvious trend during honey ripening processes. Fermentation of honey will produce ethanol, but the content of ethanol in all honey samples was relatively stable, and other studies [55,56] have also shown that the volatile components of rape honey contain ethanol, so we speculate that ethanol is an important component of natural rape honey aroma.

Nonanal, benzaldehyde monomer, and benzaldehyde dimer may be potential indicators for the identification of mature rape honey. Their content in the completely capped rape honey samples was obviously higher than that in the immature samples. The combination of antioxidant parameters and volatile components and PCA could well distinguish the rape honey samples of different maturity.

## 5. Conclusions

Herein, the physicochemical parameters, antioxidant assays, and HS-GC-IMS method were utilized to characterize honey maturity stages. With the increase in honey maturity, the moisture, sucrose and maltose content generally showed a downward trend, while the content of glucose, fructose, total protein, total phenols, total flavonoids, and proline, and the activities of diastase, sucrase, glucose oxidase, and β-glucosidase showed an overall upward tendency. In addition to moisture, the physicochemical properties of all rape honey samples agreed with current international honey standards. Hence, we think differentiating honey of different maturity stages cannot be in accordance with current honey standards. The approach of antioxidant parameters and volatile components combined with chemometrics is effective in distinguish rape honey with different maturity stages. Nonanal, benzaldehyde monomer, and benzaldehyde dimer may be the underlying indicators for the identification of mature rape honey.

## Figures and Tables

**Figure 1 foods-10-02882-f001:**
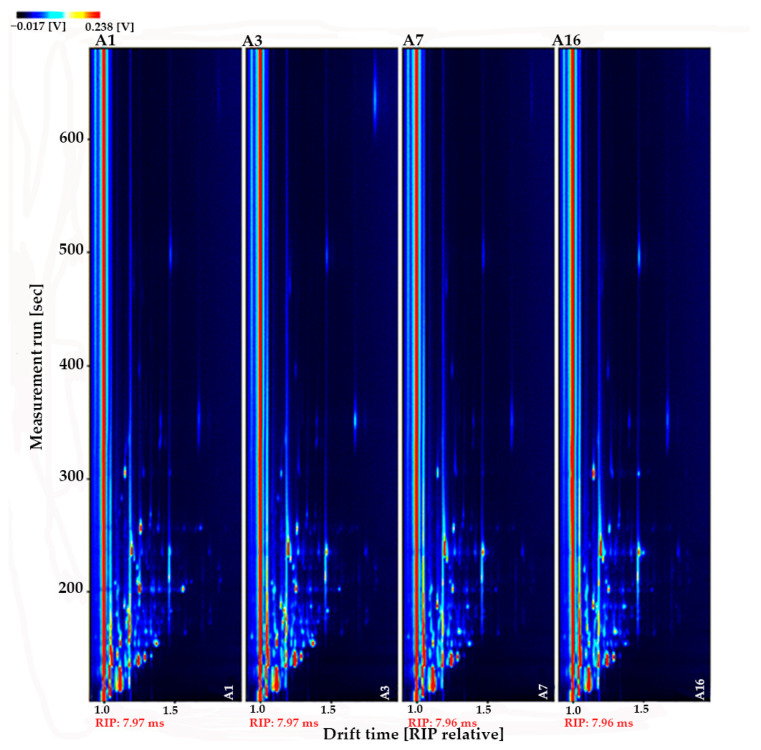
Two-dimensional spectra of volatile substances of different maturity rape honey samples (vertical view).

**Figure 2 foods-10-02882-f002:**
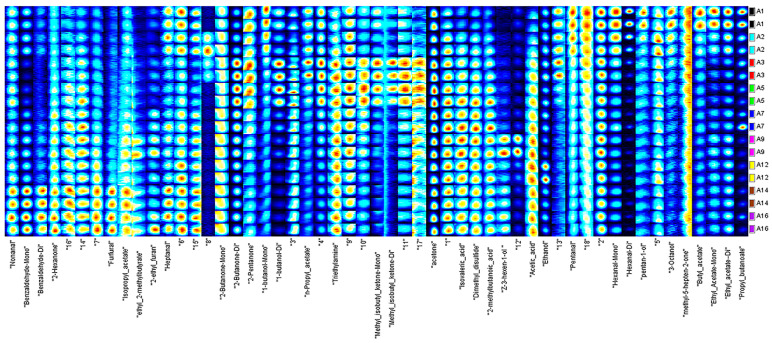
Gallery Plot fingerprint of different maturity rape honey samples.

**Figure 3 foods-10-02882-f003:**
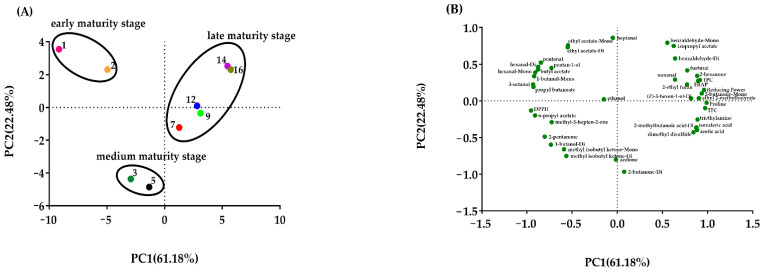
The principal component analysis (PCA) score plots (**A**) and PCA loading plots (**B**) of antioxidant and volatile components in different maturity rape honey samples. TPC: total phenolic content. TFC: total flavonoids content. FRAP: ferric reducing/antioxidant power.

**Table 1 foods-10-02882-t001:** Physicochemical indexes of different maturity rape honey samples.

Ripening Day (d)	Capping Ratio (%)	Predominant Pollen (%)	Moisture (%)	Glucose (%)	Fructose (%)	Sucrose (%)	Maltose (%)
1	0 ^g^	85.62 ± 3.6 ^ab^	26.9 ± 0.6 ^a^	32 ± 2 ^a^	28 ± 1 ^a^	0.85 ± 0.03 ^a^	1.63 ± 0.03 ^a^
2	0 ^g^	83.15 ± 2.1 ^ab^	24.9 ± 0.2 ^a^	34 ± 4 ^a^	29 ± 1 ^a^	0.78 ± 0.04 ^ab^	1.54 ± 0.03 ^b^
3	0 ^g^	88.36 ± 1.9 ^ab^	22 ± 1 ^b^	35 ± 4 ^a^	30 ± 2 ^a^	0.83 ± 0.05 ^a^	1.32 ± 0.02 ^c^
4	0 ^g^	86.15 ± 4.5 ^ab^	22 ± 1 ^bc^	36 ± 2 ^a^	30 ± 2 ^a^	0.72 ± 0.01 ^b^	1.24 ± 0.01 ^d^
5	2.3 ± 0.2 ^fg^	83.24 ± 3.2 ^ab^	21 ± 1 ^bcd^	35 ± 6 ^a^	30 ± 3 ^a^	0.60 ± 0.03 ^c^	1.16 ± 0.01 ^e^
6	9.6 ± 0.7 ^f^	85.37 ± 3.3 ^ab^	20.1 ± 0.6 ^bcd^	35 ± 6 ^a^	29 ± 2 ^a^	0.61 ± 0.04 ^c^	1.08 ± 0.02 ^f^
7	22 ± 2 ^e^	89.43 ± 1.6 ^ab^	19.9 ± 0.9 ^bcd^	35 ± 4 ^a^	30 ± 3 ^a^	0.57 ± 0.05 ^c^	1.11 ± 0.01 ^ef^
8	27 ± 4 ^e^	80.56 ± 2.8 ^b^	20.1 ± 0.5 ^bcd^	34 ± 5 ^a^	30 ± 3 ^a^	0.47 ± 0.01 ^d^	1.02 ± 0.01 ^g^
9	47 ± 3 ^d^	90.48 ± 3.1 ^a^	19.5 ± 0.8 ^cd^	35 ± 3 ^a^	31 ± 1 ^a^	0.46 ± 0.03 ^d^	0.84 ± 0.05 ^h^
10	69 ± 6 ^c^	81.34 ± 4.1 ^b^	20 ± 1 ^cd^	35.4 ± 0.8 ^a^	31 ± 2 ^a^	0.46 ± 0.01 ^d^	0.87 ± 0.01 ^h^
11	78 ± 6 ^b^	85.26 ± 2.2 ^ab^	20 ± 1 ^cd^	36 ±3 ^a^	31 ± 1 ^a^	0.45 ± 0.00 ^d^	0.88 ± 0.01 ^h^
12	85 ± 5 ^b^	88.36 ± 4.3 ^ab^	19.2 ± 0.2 ^d^	35 ± 5 ^a^	33 ± 2 ^a^	0.45 ± 0.02 ^d^	0.73 ± 0.02 ^i^
14	100.0 ± 0.0 ^a^	83.29 ± 2.7 ^ab^	18.5 ± 0.2 ^d^	37 ± 4 ^a^	35 ± 4 ^a^	0.43 ± 0.00 ^d^	0.67 ± 0.02 ^j^
16	100.0 ± 0.0 ^a^	82.47 ± 1.8 ^ab^	18.5 ± 0.7 ^d^	39 ± 1 ^a^	35 ± 2 ^a^	0.41 ± 0.03 ^d^	0.67 ± 0.02 ^j^
**Ripening** **Day** **(d)**	**Electrical** **Conductivity** **(μs/cm)**	**pH**	**Total Protein** **(mg/g)**	**Diastase** **(DN)**	**Invertase** **(mg/g·h)**	**Glucose Oxidase** **(mg/g·0.5 h)**	**β-glucosidase** **(U/g)**
1	179 ± 5 ^a^	3.72 ± 0.01 ^a^	6.7 ± 0.5 ^c^	16 ± 4 ^c^	44 ± 6 ^e^	133 ± 41 ^e^	0.85 ± 0.02 ^c^
2	171 ± 2 ^a^	3.60 ± 0.01 ^cde^	6.9 ± 0.6 ^b^^c^	19 ± 4 ^b^^c^	47 ± 5 ^e^	165 ± 15 ^d^^e^	0.89 ± 0.06 ^c^
3	162 ± 18 ^a^	3.66 ± 0.02 ^abc^	7.5 ± 0.6 ^abc^	21 ± 6 ^abc^	85 ± 9 ^de^	189 ± 42 ^cde^	0.88 ± 0.03 ^c^
4	164 ± 2 ^a^	3.69 ± 0.03 ^ab^	7.4 ± 0.6 ^abc^	20 ± 4 ^abc^	112 ± 15 ^cd^	192 ± 29 ^cde^	0.91 ± 0.02 ^c^
5	152 ± 15 ^a^	3.63 ± 0.00 ^bcd^	7.4 ± 0.6 ^abc^	21 ± 4 ^abc^	117 ± 27 ^cd^	240 ± 7 ^bcd^	0.92 ± 0.04 ^c^
6	155 ± 10 ^a^	3.55 ± 0.04 ^e^	7.9 ± 0.6 ^abc^	22 ± 3 ^abc^	123 ± 26 ^b^^cd^	252 ± 5 ^abcd^	0.94 ± 0.04 ^c^
7	159 ± 11 ^a^	3.58 ± 0.03 ^de^	8.5 ± 0.7 ^abc^	23 ± 2 ^abc^	131.3 ± 0.4 ^abcd^	280.0 ± 0.1 ^abc^	0.95 ± 0.03 ^c^
8	162 ± 14 ^a^	3.61 ± 0.03 ^cde^	8.4 ± 0.7 ^abc^	23 ± 2 ^abc^	130 ± 20 ^abcd^	260 ± 52 ^abcd^	1.11 ± 0.02 ^b^
9	162 ± 9 ^a^	3.56 ± 0.01 ^de^	8.1 ± 0.7 ^abc^	24 ± 3 ^abc^	149 ± 20 ^abc^	284 ± 15 ^abc^	1.17 ± 0.02 ^b^
10	160 ± 11 ^a^	3.54 ± 0.03 ^e^	8.4 ± 0.7 ^abc^	24.9 ± 0.3 ^abc^	162 ± 21 ^abc^	307 ± 7 ^ab^	1.12 ± 0.04 ^b^
11	166 ± 6 ^a^	3.58 ± 0.00 ^de^	8.7 ± 0.7 ^ab^	25.2 ± 0.4 ^ab^	167 ± 27 ^abc^	353 ± 66 ^a^	1.21 ± 0.03 ^b^
12	164 ± 7 ^a^	3.56 ± 0.05 ^e^	8.6 ± 0.7 ^abc^	26.06 ± 0.01 ^ab^	163 ± 31 ^abc^	352 ± 42 ^a^	1.21 ± 0.02 ^b^
14	156 ± 1 ^a^	3.59 ± 0.01 ^cde^	8.6 ± 0.7 ^abc^	29 ± 1 ^a^	180 ± 24 ^ab^	305 ± 53 ^ab^	1.40 ± 0.07 ^a^
16	161 ± 8 ^a^	3.60 ± 0.02 ^cde^	8.8 ± 0.7 ^a^	29 ± 1 ^a^	189 ± 19 ^a^	276 ± 32 ^abc^	1.41 ± 0.01 ^a^

Data represent the mean of triplicate readings ± standard deviations (SD). Different lower case letters correspond to significant differences at *p* < 0.05.

**Table 2 foods-10-02882-t002:** Antioxidant components, proline content and antioxidant activity of different maturity rape honey samples.

Ripening Day (d)	TPC (mg GAE/100 g)	TFC (mg QE/100 g)	Proline (mg/kg)	DPPH (EC50, g/mL)	FRAP (mmol Fe^2+^/100 g)	Reducing Power (mg/100 g)
1	23.8 ± 0.9 ^b^	24 ± 2 ^c^	212 ± 13 ^c^	0.63 ± 0.07 ^a^	0.74 ± 0.08 ^f^	83 ± 4 ^e^
2	24 ± 2 ^b^	26 ± 2 ^bc^	245 ± 12 ^bc^	0.58 ± 0.01 ^ab^	0.86 ± 0.09 ^f^	88 ± 6 ^de^
3	24.5 ± 0.4 ^b^	28 ± 3 ^abc^	261 ± 32 ^abc^	0.53 ± 0.03 ^bc^	1.03 ± 0.04 ^ef^	94 ± 6 ^cde^
4	25.1 ± 0.6 ^b^	29 ± 3 ^abc^	262 ± 17 ^abc^	0.51 ± 0.02 ^bcd^	1.03 ± 0.04 ^ef^	94 ± 5 ^cde^
5	26 ± 2 ^ab^	31 ± 3 ^abc^	265 ± 12 ^abc^	0.51 ± 0.02 ^bcd^	1.06 ± 0.08 ^def^	95 ± 7 ^cde^
6	27 ± 2 ^ab^	31 ± 4 ^abc^	289 ± 35 ^ab^	0.48 ± 0.00 ^cde^	1.06 ± 0.06 ^def^	95 ± 7 ^cde^
7	28 ± 2 ^ab^	32 ± 1 ^abc^	299 ± 8 ^ab^	0.48 ± 0.01 ^cdef^	1.3 ± 0.2 ^de^	98 ± 7 ^bcde^
8	27 ± 1 ^ab^	32 ± 2 ^abc^	292 ± 18 ^ab^	0.44 ± 0.04 ^def^	1.4 ± 0.1 ^cde^	100 ± 11 ^abcde^
9	27.4 ± 0.9 ^ab^	33 ± 2 ^ab^	294 ± 13 ^ab^	0.43 ± 0.00 ^efg^	1.40 ± 0.02 ^cd^	103 ± 8 ^abcd^
10	27.3 ± 0.5 ^ab^	33 ± 3 ^ab^	298 ± 10 ^ab^	0.40 ± 0.01 ^fgh^	1.7 ± 0.1 ^bc^	106 ± 2 ^abcd^
11	30 ± 2 ^ab^	34 ± 4 ^ab^	306 ± 22 ^ab^	0.36 ± 0.02 ^ghi^	1.9 ± 0.2 ^ab^	106 ± 2 ^abc^
12	30 ± 4 ^ab^	34 ± 3 ^ab^	306 ± 30 ^ab^	0.34 ± 0.03 ^hi^	2.1 ± 0.2 ^ab^	108 ± 3 ^abc^
14	32 ± 5 ^a^	33 ± 4 ^ab^	314 ± 46 ^a^	0.33 ± 0.01 ^hi^	2.2 ± 0.2 ^a^	114 ± 5 ^ab^
16	32 ± 3 ^a^	35 ± 4 ^a^	318 ± 18 ^a^	0.31 ± 0.03 ^i^	2.3 ± 0.1 ^a^	118 ± 7 ^a^

Data represent the mean of triplicate readings ± standard deviations (SD). GAE: gallic acid. QE: quercetin. TPC: total phenolic content. TFC: total flavonoids content. FRAP: ferric reducing/antioxidant power. Different lower case letters correspond to significant differences at *p* < 0.05.

**Table 3 foods-10-02882-t003:** Correlation analysis of antioxidant components and antioxidant activity.

	TPC	TFC	Proline	DPPH	FRAP	Reducing Power
TPC	1					
TFC	0.866 **	1				
Proline	0.882 **	0.971 **	1			
DPPH	−0.950 **	−0.909 **	−0.919 **	1		
FRAP	0.962 **	0.860 **	0.857 **	−0.971 **	1	
Reducing Power	0.954 **	0.915 **	0.916 **	0.963 **	0.969 **	1

** Correlation is significant at the 0.01 level (2-tailed). TPC: total phenolic content. TFC: total flavonoids content. FRAP: ferric reducing/antioxidant power.

**Table 4 foods-10-02882-t004:** Volatile components found in honey samples.

Count	Compound	RI	Rt [Sec]	Dt [RIPrel]	Peak Areas								
					1	2	3	5	7	9	12	14	16
	Aldehydes												
1	nonanal	1106.4	499.475	1.4738	166.43	172.19	177.55	186.49	163.34	167.27	182.64	236.66	226.20
2	benzaldehyde-Mono	955.9	306.492	1.1505	173.15	177.60	66.95	81.93	149.10	177.21	166.61	305.36	294.93
3	benzaldehyde-Di	954.5	305.332	1.4728	24.98	25.40	18.72	21.46	24.40	25.34	25.60	51.96	49.71
4	heptanal	892.2	260.223	1.3286	37.07	41.73	27.56	25.10	34.07	27.72	30.90	37.50	39.71
5	pentanal	687.9	166.099	1.1847	256.03	238.14	184.02	175.89	178.89	171.68	181.55	177.58	175.20
6	hexanal-Mono	784.4	203.044	1.2565	258.97	213.92	140.60	116.72	122.00	109.31	87.30	102.90	100.95
7	hexanal-Di	783.7	202.677	1.5647	194.96	135.51	74.13	59.96	61.79	56.98	45.52	52.88	56.47
8	furfural	820.5	220.276	1.0831	25.78	28.83	30.27	27.43	29.76	35.38	34.84	56.81	59.00
	Ketones												
9	2-butanone-Mono	568.1	138.666	1.0586	350.73	393.90	392.81	390.30	434.96	447.73	410.43	447.16	459.03
10	2-butanone-Di	572.2	139.512	1.2468	650.94	766.18	1655.06	1657.30	1176.66	1056.55	945.86	919.30	950.37
11	acetone	486.1	121.76	1.1171	3860.92	3169.63	4247.45	4712.79	4203.60	3731.30	3861.29	3688.94	3768.52
12	methyl isobutyl ketone-Mono	739.8	183.982	1.1704	52.64	39.69	72.90	65.62	38.10	34.43	28.91	32.09	33.98
13	methyl isobutyl ketone-Di	738.3	183.368	1.4806	28.48	24.73	39.32	40.63	23.82	21.61	19.83	16.15	20.63
14	methyl-5-hepten-2-one	986	335.12	1.181	71.42	69.15	69.81	68.41	63.21	64.41	67.12	49.12	62.91
15	2-hexanone	797	208.856	1.1858	92.45	94.66	94.18	94.67	108.63	110.44	107.73	114.47	127.39
16	2-pentanone	674.6	162.378	1.1205	62.21	51.62	67.58	62.90	45.85	43.60	33.40	35.45	36.55
	Alcohols												
17	3-octanol	984.2	333.27	1.4009	77.04	71.69	49.22	47.13	34.98	43.37	36.43	27.59	35.12
18	pentan-1-ol	759.4	191.966	1.252	81.63	63.68	57.90	63.13	61.76	59.49	57.84	60.29	60.83
19	1-butanol-Mono	653.9	157.1	1.18	150.56	143.49	101.96	97.48	97.30	93.71	87.95	84.51	81.34
20	1-butanol-Di	644.6	154.886	1.3745	145.67	96.62	191.00	172.08	77.84	64.33	48.66	40.18	41.24
21	ethanol	442.5	112.756	1.0448	599.24	691.37	665.54	510.66	575.62	399.16	991.81	512.12	518.59
	Esters												
22	propyl butanoate	887.7	257.468	1.2615	327.00	211.46	211.42	147.18	173.15	105.32	119.23	105.06	125.44
23	n-propyl acetate	709.7	172.973	1.1617	114.67	105.66	118.64	89.68	72.09	61.30	53.71	54.57	61.09
24	ethyl acetate-Mono	600.9	145.434	1.0965	127.95	99.31	76.45	73.50	82.71	93.08	89.40	86.09	90.59
25	ethyl acetate-Di	598.1	144.836	1.3381	42.49	26.59	18.19	15.61	20.97	25.38	23.06	23.30	22.51
26	(Z)-3-hexen-1-ol-Di	849.8	235.56	1.5036	7.69	14.28	18.74	25.13	28.33	69.31	36.43	40.32	48.46
27	butyl acetate	799.5	210.074	1.2371	69.23	46.52	32.97	31.44	31.72	26.79	27.48	26.21	29.40
28	isopropyl acetate	648.1	155.713	1.1573	29.98	31.83	21.21	24.16	30.23	36.10	32.61	41.06	38.14
29	ethyl 2-methylbutyrate	850.3	235.816	1.2316	17.47	20.44	28.35	31.81	34.72	58.52	38.34	49.37	49.89
	Acids												
30	2-methylbutanoic acid-Di	851.3	236.413	1.4727	124.68	170.58	179.85	203.49	213.06	215.23	205.71	197.46	212.03
31	isovaleric acid	839.9	230.272	1.2169	47.72	76.59	82.28	94.80	97.31	106.53	100.25	93.73	97.73
32	acetic acid	602.2	145.691	1.152	55.17	61.71	89.79	96.38	98.35	103.31	89.44	104.57	99.84
	Furan												
33	2-ethyl furan	684.5	165.125	1.3029	53.61	47.53	58.71	49.41	82.56	98.63	74.76	73.15	114.27
	Disulfide												
34	dimethyl disulfide	752.4	189.027	1.1471	68.17	91.41	109.12	133.03	144.69	136.71	127.03	129.69	128.27
	Amine												
35	triethylamine	686	165.546	1.2264	49.24	51.28	62.76	59.30	60.64	61.91	60.23	65.08	67.82

Data represent the mean of duplicate readings. RI: the retention index. Rt [sec]: the retention time. Dt [RIPrel]: the drift time. Di: dimer. Mono: Monomer.

## Data Availability

The data presented in this study are available within the article.

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
