# Peer review of "Investigation of the Maturity Evaluation Indicator of Honey in Natural Ripening Process: The Case of Rape Honey"

_foods, 2021, doi:10.3390/foods10112882_

Round 1

Reviewer 1 Report

SUGGESTED CORRECTIONS

  1. INTRODUCTION

Line 33: Please delete “universally”

Line 44: Do not use capital letter for citation: Zhang instead of ZHANG

Line 54: Do not use capital letter for citation: Ma instead of MA

Lines 58-59: change “sodium dodecyl sulfate polyacrylamide gel electrophoresis (SDS-PAGE)” in “SDS-PAGE (sodium dodecyl sulfate polyacrylamide gel electrophoresis)”

  1. MATERIALS AND METHODS

2.3. Physicochemical and chemical Parameters of Honey Samples

Line 130: “de Almeida-Muradian” instead of “de Alnneida-Muradian” (also in reference 18)

  1. RESULTS

3.1. Analysis of Physicochemical and Chemical Properties

Correct significant digits of data reported in the text according to Table 1.

Lines 258-260: “The results of the melissopalynological analysis revealed that rape honey samples had a percentage of pollen grains above the minimum of 45%. It showed that honey samples in our research were monofloral honeys.”  Put this sentence at the beginning of chapter 3.1.

3.2. Antioxidant Analysis and Proline Content

Correct significant digits of data reported in the text according to Table 2.

3.3. Analysis of Volatile Profiles

Figure 1: please enlarge the figure to the full width of the page so that it can be seen better.

Line 346: “With” not capital letter

Line 346: “(Z)-3-hexen-1-ol dimer” instead of“z-3-hexen-1-ol dimer” with capital letter for Z

Lines 370-371: “(Z)-3-hexen-1-ol dimer” instead of“z-3-hexen-1-ol di” with capital letter for Z

  1. DISCUSSION

Lines 389-393: Concerning the sentence, “In our study, with the exception of small fluctuations (20.13%) on the 8th day, the water content of the samples were all lower than 20% from the 7th day. The slight increase in water content of the sample on the 8th day may be due to the poor flow of necter the previous day, and fresh honey was added to the cell after the bees had eaten part of the honey.”, it can be noted that using the correct significant digits, the water content of the samples were all lower than 20% only from the 12th day. I suggest changing the sentence to "In our study the water content of the samples became 20% by day 7 and only decreased further by day 12."

Line 407: Add “(data not shown)” after “from 0.83 to 0.93

Line 411: “relevant” instead of “relevanted”

Lines 424-428: Correct significant digits of data reported in the text according to Table 1.

Line 450: Correct significant digits of data reported in the text according to Table 2.

Lines 451-452: change “The proline content in rape honey was 142-466 mg/kg, in consistent with the previous study [47]” in “The proline content in rape honey was consistent with the previous study (142-466 mg/kg) [47]”

Author Response

Line 33: Please delete “universally”

Response: Thanks. Done as suggested.

Line 44: Do not use capital letter for citation: Zhang instead of ZHANG

Response: Thanks. Done as suggested.

Line 54: Do not use capital letter for citation: Ma instead of MA

Response: Thanks. Done as suggested.

Lines 58-59: change “sodium dodecyl sulfate polyacrylamide gel electrophoresis (SDS-PAGE)” in “SDS-PAGE (sodium dodecyl sulfate polyacrylamide gel electrophoresis)”

Response: Thanks. Done as suggested.

MATERIALS AND METHODS

2.3. Physicochemical and chemical Parameters of Honey Samples

Response: Thanks. Done as suggested.

Line 130: “de Almeida-Muradian” instead of “de Alnneida-Muradian” (also in reference 18)

Response: Thanks. Done as suggested.

RESULTS

3.1. Analysis of Physicochemical and Chemical Properties

Correct significant digits of data reported in the text according to Table 1.

Response: Thanks. Done as suggested.

Lines 258-260: “The results of the melissopalynological analysis revealed that rape honey samples had a percentage of pollen grains above the minimum of 45%. It showed that honey samples in our research were monofloral honeys.” Put this sentence at the beginning of chapter 3.1.

Response: Thanks. Done as suggested.

3.2. Antioxidant Analysis and Proline Content

Correct significant digits of data reported in the text according to Table 2.

Response: Thanks. Done as suggested.

3.3. Analysis of Volatile Profiles

Figure 1: please enlarge the figure to the full width of the page so that it can be seen better.

Response: Thanks. Done as suggested.

Line 346: “With” not capital letter

Response: Thanks. Done as suggested.

Line 346: “(Z)-3-hexen-1-ol dimer” instead of“z-3-hexen-1-ol dimer” with capital letter for Z

Response: Thanks. Done as suggested.

Lines 370-371: “(Z)-3-hexen-1-ol dimer” instead of“z-3-hexen-1-ol di” with capital letter for Z

Response: Thanks. Done as suggested.

DISCUSSION

Lines 389-393: Concerning the sentence, “In our study, with the exception of small fluctuations (20.13%) on the 8th day, the water content of the samples were all lower than 20% from the 7th day. The slight increase in water content of the sample on the 8th day may be due to the poor flow of necter the previous day, and fresh honey was added to the cell after the bees had eaten part of the honey.”, it can be noted that using the correct significant digits, the water content of the samples were all lower than 20% only from the 12th day. I suggest changing the sentence to "In our study the water content of the samples became 20% by day 7 and only decreased further by day 12."

Response: Thanks. Done as suggested.

Line 407: Add “(data not shown)” after “from 0.83 to 0.93”

Response: Thanks. Done as suggested.

Line 411: “relevant” instead of “relevanted”

Response: Thanks. Done as suggested.

Lines 424-428: Correct significant digits of data reported in the text according to Table 1.

Response: Thanks. Done as suggested.

Line 450: Correct significant digits of data reported in the text according to Table 2.

Response: Thanks. Done as suggested.

Lines 451-452: change “The proline content in rape honey was 142-466 mg/kg, in consistent with the previous study [47]” in “The proline content in rape honey was consistent with the previous study (142-466 mg/kg) [47]”

Response: Thanks. Done as suggested.

Reviewer 2 Report

Thank you very much for your revision. It is good but still is necessary some minor changes,

Line 54, Please change MA by Ma.

Line 130, Please change Alnneida-Muradian by Almeida-Muradian

Line 143, The correct citation is Bentabol et al. or Bentabol-Manzanares et al., Please revise the citation in:

https://doi.org/10.1016/j.lwt.2013.09.024

Table 1, the mean values of the variables for each sample should have the same number of decimals (two or one).

Line 315, Cluster analysis, if the figure is maintained, a comment should be included to explain it.

Line 332, please include and space after databases.

Table 4. A comment justifying why data of samples taken on days 4, 6, 8 and 10 are not included is mandatory.

PCA results, a more detailed explanation of figure 3B should be very interesting, especially concerning the degree of ripening.

Author Response

Thank you very much for your revision. It is good but still is necessary some minor changes,

Line 54, Please change MA by Ma.

Response: Thanks. Done as suggested.

Line 130, Please change Alnneida-Muradian by Almeida-Muradian

Response: Thanks. Done as suggested.

Line 143, The correct citation is Bentabol et al. or Bentabol-Manzanares et al., Please revise the citation in:

https://doi.org/10.1016/j.lwt.2013.09.024

Response: Thanks. Done as suggested.

Table 1, the mean values of the variables for each sample should have the same number of decimals (two or one).

Response: We sincerely appreciate the reviewer’s careful review of our manuscript. We revised Table 1 and Table 2 based on another reviewer’s suggestion, which data reported on the two tables have too much significant digits. Therefore, the decimal number of the mean values of the variables for each sample is required by another reviewer. For example, for Moisture: 26.9 ± 0.6 instead of 26.90 ± 0.64 , for Fructose: 28 ± 1 instead of 28.01 ± 1.49.

Line 315, Cluster analysis, if the figure is maintained, a comment should be included to explain it.

Response: Thanks. We have removed the Cluster analysis.

Line 332, please include and space after databases.

Response: Thanks. Done as suggested.

Table 4. A comment justifying why data of samples taken on days 4, 6, 8 and 10 are not included is mandatory.

Response: Thanks. In chapter 2.6, we have shown that we selected samples with more significant physicochemical properties and antioxidant capacity, that is, samples on days 1, 2, 3, 5, 7, 9, 12, 14, and 16 for the determination of volatile constituents.

PCA results, a more detailed explanation of figure 3B should be very interesting, especially concerning the degree of ripening.

Response: Thanks. We have rewritten the PCA results in “The PCA results of 41 indicators (antioxidant and 35 of identified volatile components) for 9 honey samples at different maturity stages are shown in Figure 3. The two principal components explained 83.66% of the total variance, the first principal component (PC1) explained 61.18%, and the second principal component (PC2) explained 22.48%. As shown in the PCA score plot (Figure 3A), the samples studied were discriminated into three different groups which corresponded with their ripening stages. 1-day and 2-day honey samples were scattered in II quadrant (early maturity stage). 3-day and 5-day honey samples were dispersed in the circle of medium maturity stage (in â…¢ quadrant). The honey samples over 7 days were classified as late maturity stage. The score plot (Figure 3B) indicates the position of the 41 indicators. The samples in early maturity stage seemed characterised by most alcohols (3-octanol, pentan-1-ol, 1-butanol-monomer), aldehydes (hexanal dimer, hexanal monomer, pentanal) and butyl acetate (Figure 3B). Methyl isobutyl ketone (dimer and monomer) dominated in the samples of medium maturity stage. The location of honey samples at the late maturity stage was positively correlated with antioxidant parameters (TPC, TFC, FRAP, reducing power), benzaldehyde (dimer and monomer), and isopropyl acetate, which indicated these samples had relatively high concent of antioxidant components and better antioxidant capacity”.

This manuscript is a resubmission of an earlier submission. The following is a list of the peer review reports and author responses from that submission.

Round 1

Reviewer 1 Report

Investigation of the maturity evaluation indicator of honey in 2 natural ripening process: the case of rape honey

The paper reports a study to find potential maturity indicators of rape honey, evaluating physicochemical parameters, antioxidants indicators and volatile components.

The work features enough novelty, and it could be of interest for honey industry for detection of honey adulteration.

The manuscript requires some language editing; please check the English meaning of all the manuscript also with the help of a mother language speaking.

The manuscript needs some minor revisions.

Some of the references refer to papers written in Chinese (8, 12, 16, 22, 26) or unobtainable (5, 26) on the web and therefore not readily available to readers.

Do not use capital letter for citation throughout all the text (see for example ZHANG pag. 1 line 42)

I suggest adding a statistical comparison procedure on all the physicochemical and antioxidant measured data (Tukey's honest significance test for example) to find means that are significantly different from each other throughout the maturity stage.

I suggest performing a multivariate statistical analysis (PCA or CA) with both antioxidant and volatile components data together to separate samples with different maturity stage.

Data reported on Table 1 and 2 have too much significant digits. Please correct.

SUGGESTED CORRECTIONS

  1. INTRODUCTION

Line 56: Please specify what the abbreviation SDS-PAGE refer to

Lines 57-58: The sentence “Similarly, mature honey contains phenolic composition than immature honey ….” is not clear.  Maybe the word “different” is missing before “phenolic composition” ? Please write the sentence in a clearer way

Line 76: Please specify what the abbreviation °Bé refer to

  1. MATERIALS AND METHODS

Lines 118-119:For the electrical conductivity measurement, the 20% (w/v) honey solution was determined by the conductometer at 20 °C.” Please write the sentence in a clearer way.

Lines 123-124:as the number of mL of per g honey hydrolyzed 1% starch in 1 h, called the diastase number (DN)” Please write the sentence in a clearer way.

Lines 131-133:The principle is that when the substrate (4-nitrophenyl-β-D-glucopyranoside) is hydrolyzed, the 4-nitrophenol released can be calculated by a microplate reader.

Please describe the method and not the method principle. The reference is not easily available, and it is written in Chinese.

Lines 136-142:Using a pipette to transfer 1 mL each of the solution before and after conversion, 2 mL 3,5-dinitrosalicylic acid solution was added. After heating 3 min in a boiling water and cooling in a running water for 5 min, the solution was adjusted with distilled water to 25 mL and shook evenly. The absorbance was measured at a wavelength of 540 nm. The invertase activity refers to the number of mg of invertase contained in per g of honey that can convert sucrose into monosaccharides within 1 h at 45 °C.

Please write the sentence in a clearer way.

  1. RESULTS

3.1. Analysis of Physicochemical and chemical Properties

Line 232: How can you state that the decrease is “significant” if you do not perform a statistical comparison test (Tukey test for example) ? Please compare statistically results for physicochemical parameter.

Lines 241-242:the electrical conductivity in honey was in the range 152.40-178.60 μS/cm.

Add “in the range”

Lines 245-247: change the “big point •” in units with a smaller one.

Table 1: Data reported on Table 1 have too much significant digits. Please correct.

For example, for Moisture: 26.9 ± 0.6 instead of 26.90 ± 0.64

For example, for Fructose: 28 ± 1 instead of 28.01 ± 1.49

I suggest, for each measured parameter, to compare means to find if are significantly different from each other throughout the maturity stage with Tukey's honest significance test.

3.2. Antioxidant Analysis and Proline Content

Line 253: add “and proline content” in the Subtitle 3.2 because proline is correlated to the antioxidant activity but is not an antioxidant.

Line 255: substitute “with” with “at”.

add “and proline content” in the Subtitle 3.2 because proline is correlated to the antioxidant activity but is not an antioxidant.

Table 2: Data reported on Table 2 have too much significant digits. Please correct.

For example, for TPC: 23.8 ± 0.9 instead of 23.77 ± 0.89

For example, for Proline: 212 ± 13 instead of 211.75 ± 13.13

I suggest, for each measured parameter, to compare means to find if are significantly different from each other throughout the maturity stage with Tukey's honest significance test.

Lines 281-287: Since a good correlation has been found between all the parameters measured to evaluate the antioxidant activity, in my opinion it is completely useless to analyze these data by means of a multivariate cluster analysis. Perhaps the treatment with a Cluster analysis of all the antioxidant and physicochemical indexes should have more sense.

In addition the analysis is not adequately described (method, distance, all the antioxidant data ?)

Please eliminate or change Cluster Analysis treatment.

Table 4: Please correct the formatting of the table: some of the numbers wrap down.

Figure 2: The Figure contain the results obtained for all the analysed samples, but it is impossible to see and understand difference between samples. My suggestion is to report only two dimensional spectra  for early (A1), medium (A3, A8) and final (A16) maturity stage.

Line 308: “with the brewing time went” I can't understand this expression.

Figure 4: Please give more detail on the PCA analysis eg. the data used (probably you don’t use means but the single result for each sample because in the graph you had put two point for each sample), and eigenvectors (that can give you an indication about the volatile compounds that discriminate between samples).

In addition it should be maybe appropriate to encircle with ellipse not the results obtained for each sample (A1 and A1, then A3 and A3, …..) but results obtained for early maturity stage samples  (A1 and A2), medium maturity stage (A3 and A5) and late maturity stage (A7, A9, A12, A14, A16).

In addition, a suggestion …..

The CA applied on antioxidant data gave you a result similar to that obtained with PCA analysis on volatile components. Do you tried to treat with PCA analysis antioxidant and volatile components data together ? Probably you will find a good separation between the samples with different maturity stage.

  1. DISCUSSION

Lines 344-347:The sum of fructose and glucose content of all honey samples in this study was in accordance with the Codex Alimentarius [1] and European standards [2] that the content of monosaccharide is not less than 60%.” Please write the sentence in a clearer way.

Lines 359-360: Eliminate “of MA et al.

Lines 369, 371, 372, 373, 379: change the “big point •” in units with a smaller one.

Line 389: Correct “srudy

Lines 396-403: Eliminate all the data concerning the correlation reported in the literature. It is sufficient to indicate that the results are in agreement with the literature and to indicate the references.

Lines 403-405: Eliminate this sentence together with CA treatment in the RESULTS.

Lines 418-420:Acetone is the substance with the highest content of volatile components in rape honey [55], which was in good agreement with our results.” Please write the sentence in a clearer way.

Lines 423-425: See my comments in the results section about PCA.

REFERENCES

Some of the references refer to papers written in Chinese (8, 12, 16, 22, 26) or unobtainable (5, 26) on the web and therefore not readily available to readers.

Some of the references contain some error, in particular:

  2. https://doi.org/10.3390/molecules24142674 Eliminate the final full stop
  3. Add the DOI: 10.13995/j.cnki.11-1802/ts.015049
  4. Add the issue 41(22); add the WEB link (the DOI doesn’t work): https://caod.oriprobe.com/articles/60097448/Determination_of_Volatile_Components_in_Buckwheat_.htm
  5. Correct: A.O.A.C. (1990) Official Methods of Analysis. 15th Edition, Association of Official Analytical Chemist, Washington DC.
  6. Add the issue 31(19); add the WEB link (the DOI doesn’t work): https://caod.oriprobe.com/articles/38055374/feng_mi_zhong____pu_tao_tang_zuo_mei_huo_xing_ce_d.htm
  7. Correct the DOI (there is a full stop before j): https://doi.org/10.1016/j.lwt.2018.01.016

Reviewer 2 Report

The manuscript describes differences in honey samples during the ripening process. The authors have interesting data about this process in rape honey samples, but the document needs major changes.

Comments:
Line 33 to 38, It is not clear what the authors want to say in these lines. The harvest of honey every 2 or 3 days is not a common practice in beekeeping around the world to produce honey. To make more concise the explanation, common techniques to produce honey in the area where the samples were taken should be described. Then sentence in Line 35-36 should be explained because if immature honey is difficult to manage and the management is more expensive, harvesting honey every 2 or 3 days is not a favourable practice for beekeepers. Furthermore, if rape honey is obtained in a short period (fifteen days), it should be explained, and the consequences discussed together with the results in Discussion section. 
Line 68, Although some specific volatile compounds can be related to the botanical origin of honey. This statement is too categorical since honey samples of different botanical origins share volatile compounds. I suggest authors rewrite the sentence explaining the origin of volatile compounds in honey sample: from plants, from honey bees, honey ripening…
The use of brewing and maturation is common in food processing industries but not in honey production in hives. I suggest the use of “ripening” for the “maturation” of honey and avoid the use of brewing.
Lines 82-86, The objective of the study marked in the abstract (lines 11-14) is not the same described in the last paragraph of the introduction. Both should be in accordance.
Lines 89-96, The experimental design to take the samples is one of the worst aspects of the paper. In the present form, this part is not valid because is impossible to understand how the samples were taken. It is necessary to detail more when the samples were taken from the hives. For example, it would be interesting to include the date of sampling for the 14 samples.
Authors confirm that honey samples are from rape plants, but there is no evidence that during the honey ripening other honey flow can be introduced in hives. So that a pollen analysis of honey samples would be very interesting to confirm the origin and to know if other plants can affect honey properties.
Lines 103-104, DPPH and TPTZ, are common products in honey analyses but the full name of both should be written. The same for FRAP.
Lines 115, change arguments for parameters.
The methodology of physicochemical analysis should be based on the original descriptions or on honey analysis methods. The reference PASCUAL-MATE et al. [17]. doesn’t describe the methods. The reference MANZANARES et al. [19], should be Bentabol et al.
References should be without capital letters.
Line 211, 99.99
About the statistical analysis, both Cluster Analysis and PCA analysis are used to interpret a data matrix with many cases and many variables. So that in my opinion, these types of treatments are not appropriate for the interpretation of the results of the samples (14) that are studied in this work. However, it would be very interesting to perform a statistical analysis showing differences between samples capped a (statistical significance of the differences). They can be marked in Tables 1 and 2.
Table 1, the information about what is Columns1 and 2, should be included in the Sampling methodology (Material and Methods). Change “Brewing” for “Ripening”
Table 3 is not necessary, because it is known that antioxidant components are highly correlated. It doesn’t provide important information for the aim of the paper: look for changes during honey ripening.
Figure 2. It is nice, but it does not provide any relevant information to know the evolution of the compounds studied during the maturation of honey. However, a more detailed description of Table 4 is necessary, especially the role of some compounds as Ethanol.
Table 4. Some lines havemoved.
“Qualitative results of volatile components in different maturity rape honey samples”, Maybe “Volatile components found in honey samples”
There are some mistakes: for example, Line 266, “continues”, Line 384, 385, moisture, line 389 study,
The discussion section should be rewritten focused on the changes during honey ripening and a more depth explanation of the data obtained during the study. For example, identify clearly what are the physicochemical parameters that presented significant changes and differences between capped and uncapped honey.
 4.3 subsection should show which are the most representative volatile compounds in this honey type and how they changed during ripening. And don’t forget to talk about ethanol.

Finally, the list of references should be checked, there are some mistakes and at least one of them is repeated (Manyi-Loh, C.E.; Ndip, R.N.; Clarke, A.M. Int. J. Mol. Sci. 2011.)

I hope my comments help authors to improve their work and encourage them to revise the paper.